# Dynamics of trachoma infection in West Africa revealed by a hidden state model

Jake Carson [1]*, Thomas Crellen[2,3,4], Anna Borlase[5], Joaquin M. Prada[6], Robin Bailey[7], T. Déirdre Hollingsworth[3,4], Simon E. F. Spencer[8]

**1** Mathematics Institute, University of Warwick, Coventry, United Kingdom, **2** Saw Swee Hock School of Public Health, National University of Singapore, Singapore, Singapore, **3** Nuffield Department of Medicine, Centre for Tropical Medicine and Global Health, University of Oxford, Oxford, United Kingdom, **4** Big Data Institute, Li Ka Shing Centre for Health Information and Discovery, University of Oxford, Oxford, United Kingdom, **5** Department of Biology, University of Oxford, Oxford, United Kingdom, **6** Faculty of Health and Medical Sciences, University of Surrey, Surrey, United Kingdom, **7** Faculty of Infectious & Tropical Disease, London School of Hygiene & Tropical Medicine, London, United Kingdom, **8** Department of Statistics, University of Warwick, Coventry, United Kingdom

* Jake.Carson@warwick.ac.uk

## Abstract

Trachoma is estimated to be the leading infectious cause of blindness globally, predominantly affecting low-income populations with poor sanitation and hygiene. Over a decade of mass drug administration with antibiotics has led to substantial progress in control and elimination, but hotspots remain where infection persists or rebounds following mass drug administration for reasons that remain unclear. Transmission modelling is a key component of understanding these dynamics, but the complex dynamics of infection and reinfection with *Chlamydia trachomatis* are challenging to infer from cross–sectional surveys. Here, we analyze longitudinal data collected over six months in 1991 using multiple diagnostics from two villages in The Gambia by developing and fitting a Bayesian epidemiological model that classifies individuals into disease states at each time point using a forward-filtering backward-sampling algorithm. We find that infection risk is clustered within households and the weekly probability of transmission within a shared room is 40–fold higher than in a shared village. Infected children are estimated to contribute disproportionately to transmission, accounting for 70–90% of the force of infection within the observed period. We estimate the basic reproduction number, $R_0$, to be 2.2 by simulation and find that the distribution of secondary cases per individual is less aggregated than for other directly-transmitted pathogens. We further quantify heterogeneity in predisposition to becoming infected and estimate the sensitivity and specificity for PCR, antigen detection tests, and clinical examinations. Our study uncovers the natural history of trachoma infection, with implications for simulating pathogen dynamics and designing interventions to halt transmission and prevent avoidable cases of blindness.

Data availability statement: The inference code and anonamised data are available at DOI: 10.5281/zenodo.17600334.

Funding: The authors gratefully acknowledge funding of the NTD Modelling Consortium by the Bill and Melinda Gates Foundation (grant INV-030046). SEFS, JC and TDH gratefully acknowledge funding from MRC grant MR/P026400/1. The funders had no role in study design, data collection and analysis, decision to publish, or preparation of the manuscript.

Competing interests: The authors have declared that no competing interests exist.

## Author summary

Trachoma is an infectious disease that can lead to blindness through repeated infections over time. Understanding trachoma transmission is important for designing surveys, evaluating the impact of different intervention strategies, and allocating resources for control programmes. Here, we infer transmission properties by analysing data from two villages in The Gambia, in which the same cohort of individuals were followed for six months. By developing and fitting an individual-level model to the 1410 individuals, we derive the impacts of household structure and age on trachoma transmission. Our analysis finds that infection risk is strongly impacted by household structure, with transmission between individuals sharing a room being 40 times higher than between individuals sharing only a village. We also find that transmission is dominated by children, who contribute over 70% of the force of infection over the study period. We further quantify differences in predisposition to infection between individuals. Finally, we determine the error rates of PCR, antigen detection tests, and clinical examinations, which were used during the study.

## Introduction

Trachoma is an infectious cause of blindness caused by ocular strains of the gram-negative bacterium *Chlamydia trachomatis* and is classified as a neglected tropical disease [1]. Repeated infection leads to chronic inflammation of the conjunctiva and to scarring of the inner eyelid (trachomatous scarring; TS) [2]. Over time this scar tissue contracts, which can cause distortion of the eyelids, entropion (inward rotation of the eyelid), leading to contact between the eyelashes and the surface of the eye (trachomatous trichiasis; TT). This is an acutely painful condition, and can ultimately lead to corneal opacity, visual impairment and irreversible blindness. Transmission is mediated person-to-person by ocular or nasal discharge via hands, fomites (e.g., bedding) and eye–seeking flies [3]. As of March 2026, trachoma is endemic in 30 countries and is responsible for blindness or visual impairment in approximately 1.9 million people, with the majority of cases in African countries [4]. Ambitious targets have been set for the control of trachoma worldwide, with all remaining endemic countries aiming for "elimination as a public health problem" by 2030, which requires a prevalence of trachomatous inflammation–follicular (TF; a marker of current or recent infection) <5% in children 1–9 years old, and a prevalence of TT < 0.2% in people 15 years old and over [5]. Mass treatment with the broad-spectrum antibiotic azithromycin is the main tool used to clear infection and prevent onward transmission [6,7]. Other disease control strategies include facial cleanliness, environmental improvement to reduce fly density, and the provision of clean water [8]. Surgical interventions can treat trichiasis [3], with evidence that health systems can manage TT cases being a third criterion for elimination as a public health problem.

Transmission modelling plays an important role within trachoma elimination programmes for survey design, evaluating the impact of interventions, and allocating future resources [7,9,10]. Understanding the mechanisms of pathogen transmission underpin these large-scale predictive models [11]. Key aspects of trachoma transmission that are important to quantify from epidemiological data include i) how the number of times a person has been infected, and their age, affects the course of subsequent infections, e.g., the observation of shorter infectious periods and lower bacterial load in older patients [12], ii) heterogeneity in susceptibility [13] and iii) declining transmission over time in untreated regions [14]. However, most data take the form of cross-sectional surveys, rather than repeated observations, making it challenging to infer hidden or partially observed epidemiological processes.

Here we fit an individual-level stochastic transmission model to detailed longitudinal data collected from endemic communities in West Africa in the early 1990s. This dataset, observed prior to the demonstration of the efficacy of single-dose azithromycin, provides a unique resource for modelling the dynamics of trachoma infection because no intervention was applied for the six month observation period [13,15]. Previous analyses of this cohort have focused on the duration of infection in adults and children, diagnostic efficacy and household transmission [12,13,16,17]. We are the first to infer the epidemiological state of individuals, characterize heterogeneity in susceptibility, and quantify the contribution of adults and children to transmission. The model structure and parameters estimated here will inform a predictive model of trachoma transmission throughout the African continent to inform disease control efforts and policy planning [18].

## Materials and methods

### Ethics statement

The data used in this study derive from a previously conducted cohort study, which received ethical approval from the Joint Gambia Government and Medical Research Council Ethics Committee (SCC 508) [16]. In the original study, formal verbal informed consent was obtained from all participants, with consent witnessed and documented in accordance with standard procedures at the time. No new data collection involving human participants was undertaken for the present study.

### Longitudinal data

We use data from a cohort study in two villages in The Gambia between April and November 1991, Jali (J; population 893) in Kiang West district and Berending (B; population 517) in Kombo South district, which lie approximately 90 km apart [13,15]. The dataset includes trachoma diagnostics and a clinical eye examination at baseline; the age of the surveyed participants; the residential structure (compounds within villages, rooms within compounds); and, for people included in the cohort, subsequent diagnostics taken every two weeks and a further clinical examination at the end of the study period. The village structure and sample size are summarised in Table 1.

At the start of the observed period, the eyes of all available individuals were clinically examined, and conjunctival swabs taken for antigen detection tests and PCR targeting a plasmid sequence. During the clinical examinations signs

**Table 1. Summary of the household data used in this study, showing the number of residential compounds, rooms and occupancy within the two study villages (J and B) in The Gambia.**

| Village | Compounds | Rooms | Population[1] | Per compound[2] | Per room[2] | Children[3] | Cohort[4] |
|---------|-----------|-------|------------|--------------|----------|----------|--------|
| Jali (J) | 49 | 265 | 893 | 18 (4–79) | 3.1 (1–17) | 475 (53%) | 188 |
| Berending (B) | 38 | 114 | 517 | 14 (1–52) | 4.1 (1–19) | 282 (55%) | 68 |

[1]Sample size at baseline, representing the total population of the village. [2]Mean number of people and range. [3]Children 15 years and under in the total population. [4]Sample size recruited for longitudinal observations over 25 weeks.

of trachoma were graded according to degree of inflammation, presence of follicles, and sequelae (including conjunctival scarring, entropion and corneal opacity). The PCR and antigen detection tests indicate the presence or absence of *Chlamydia trachomatis* for each individual.

Following the baseline survey, 68 individuals from Berending and 188 individuals from Jali were surveyed longitudinally over a six month period. Participants were included from twenty randomly selected households, based on the criteria of having at least one confirmed case of active trachoma and at least four household members without active trachoma [13]. Follow-ups occurred every two weeks, in which participants were clinically examined for signs of trachoma and swabs taken for antigen detection tests. There are some instances of missing data due to absences during the survey and diagnostics failing to return readable results. On the final week all available individuals underwent a second clinical eye examination. Note that we exclude data from individuals under 1 year of age as clinical signs of trachoma are considered unreliable in under 1s. These individuals are still included in the model described in the next section, but are treated as having missing data.

## Trachoma transmission model

We adapted previously published transmission frameworks as an individual-based stochastic model, informed by our understanding of the natural history of trachoma infection [9,12,16,19]. In endemic regions, people undergo repeated trachoma infections throughout their lifetime and those with past infections are known to have shorter infection clearance and disease recovery periods, possibly due to an acquired immune response [12]. We consider four possible epidemiological states in our model; susceptible ($S$), infectious ($I$), infectious and diseased ($ID$), and diseased ($D$). Here, 'diseased' refers to TF, from which individuals may recover in the absence of reinfection. Individuals transition $S \to I$ after exposure to *C. trachomatis*; $I \to ID$ following an incubation period; $ID \to D$ when the infection is cleared; and finally $D \to S$ after a recovery period. While in state $D$ individuals may also be reinfected, albeit with a modified level of susceptibility given by the parameter $\rho$. To account for a different susceptibility and duration of infectiousness we consider an individual's first infection separately from all subsequent infections and therefore denote naïve states as $S_0$, $I_0$, $ID_0$, $D_0$, and non-naïve states $S_1$, $I_1$, $ID_1$, $D_1$. The model structure is shown in Fig 1.

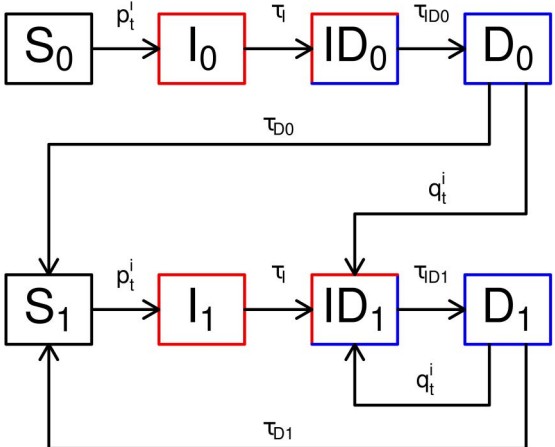

**Fig 1. Structure of the trachoma transmission model.** Individuals progress through susceptible ($S$), infectious ($I$), infectious and diseased ($ID$), and diseased states ($D$). We distinguish between first infection (0) and any later infections (1). Transmission is represented by the $p$ and $q$ parameters, and $\tau$ are durations.

Our epidemiological framework represents a simplification of previous differential equation models, which used distinct states for each subsequent infection with *C. trachomatis*, e.g., $I_n$ where $n \in \{0, 1, 2, ..., 100\}$, referred to as a "ladder of infection" [9]. One reason for simplifying this to naïve ($I_0$) and non-naïve ($I_1$) is that it is impossible to identify an individual's exact number of previous infections within the six month observation period, particularly as many rungs of the ladder exhibit similar dynamics. Our stochastic model also introduces heterogeneity to the durations in each state, removing the need for a ladder to encompass different values.

For individuals in state $S$ (taken to mean either $S_0$ or $S_1$) we define the probability of transitioning to state $I$ in a single time step (one week). Let $v_i$, $c_i$, $r_i$ denote the village, compound, and room of person $i$ respectively. Furthermore, let $\mathcal{V}_t^{v_i}$, $\mathcal{C}_t^{c_i}$, $\mathcal{R}_t^{r_i}$ be the number of infectious people (in states $I_0$, $I_1$, $ID_0$, $ID_1$, assumed equally infectious) at time $t$ in village $v_i$, compound $c_i$, and room $r_i$ respectively. We define

$$p_t^i = 1 - \exp\{-(\beta_1^{v_i}\mathcal{V}_t^{v_i} + \beta_2^{v_i}\mathcal{C}_t^{c_i} + \beta_3^{v_i}\mathcal{R}_t^{r_i})\}$$

(1)

as the probability of transitioning $S \to I$ between time $t$ and time $t+1$. This probability is derived from a Poisson process and represents the probability of at least one event occurring in unit time with rate $\beta_1^{v_i}\mathcal{V}_t^{v_i} + \beta_2^{v_i}\mathcal{C}_t^{c_i} + \beta_3^{v_i}\mathcal{R}_t^{r_i}$. The parameters $\beta_1^{v_i}$, $\beta_2^{v_i}$, and $\beta_3^{v_i}$ are coefficients that determine the infectiousness between individuals that are known to share a village, compound, or room. Individuals that share a room must share a compound and individuals in the same compound share a village. Values of $\beta_2^{v_i} > 0$ indicate that infectious individuals sharing a compound have a greater contribution to the infection pressure than individuals who only share a village. Values of $\beta_3^{v_i} > 0$ indicate that infectious individuals sharing a room have a greater contribution to the infection pressure than individuals who only share a compound and/or village. Separate parameters are used for the two villages (denoted by the superscript $v_i$), and we assume no mixing between them.

For individuals in state $I$ we define the duration spent in this state before progressing to state $ID$. We denote the duration as $\tau_I$ and the mean duration as $\mu_I$. Then $\tau_I$ is distributed according to a translated negative binomial distribution with mean $\mu_I - 1$ and size 2:

$$\tau_I \sim 1 + \text{NB}(\mu_I - 1, 2).$$

(2)

An equivalent formulation would be to duplicate state $I$ and model transitions through two sequential infectious compartments with geometric durations, yielding a Markov structure [20]. However, because the inference algorithm we employ scales quadratically with the number of model states [21], this formulation would substantially increase computational cost. We therefore adopt the negative binomial specification, where $\text{NB}(m, n)$ is defined by the following probability mass function:

$$p(x \mid m, n) = \frac{\Gamma(x + n)}{\Gamma(n)x!} \left(\frac{n}{n + m}\right)^n \left(\frac{m}{n + m}\right)^x, \quad x = 0, 1, 2, ...,$$

(3)

where $m$ is the mean and $n$ is the size parameter of the negative binomial distribution. By translating the negative binomial distribution we ensure that the duration is at least one week. Negative binomial distributions are commonly employed in infectious disease modelling to give unimodal probability distributions with a mode bigger than zero. Here, the mode of $\tau_I$ is $\lfloor \mu_I/2 \rfloor$. We make no distinction for individuals being naïve or non-naïve, and the parameters are assumed to apply to both villages.

For individuals in state $ID$ we define the duration spent in this state before progressing to state $D$. As with state $I$, we model these durations using translated negative binomial distributions. However, here a distinction is made between naïve and non-naïve individuals, with the mean duration being $\mu_{ID0}$ for naïve individuals, and $\mu_{ID1}$ for non-naïve individuals:

$$\tau_{ID0} \sim 1 + \text{NB}(\mu_{ID0} - 1, 2),$$
$$\tau_{ID1} \sim 1 + \text{NB}(\mu_{ID1} - 1, 2). \tag{4}$$

Individuals in state $D$ can either progress to state $S$ via recovery, or to state $ID$ via reinfection. The duration spent in state $D$ before progressing to state $S$ follows a translated negative binomial distribution, with the mean duration being $\mu_{D0}$ for naïve individuals, and $\mu_{D1}$ for non-naïve individuals:

$$\tau_{D0} \sim 1 + \text{NB}(\mu_{D0} - 1, 2),$$
$$\tau_{D1} \sim 1 + \text{NB}(\mu_{D1} - 1, 2). \tag{5}$$

Over these durations, individuals may become reinfected with a modified infection probability. Specifically, the infection rate is multiplied by a parameter $\rho$, which can make infection more ($\rho > 1$) or less ($\rho < 1$) likely than in the $S$ state:

$$q_t^i = 1 - \exp\{-\rho \left(\beta_1^{v_i} \mathcal{V}_t^{v_i} + \beta_2^{v_i} \mathcal{C}_t^{c_i} + \beta_3^{v_i} \mathcal{R}_t^{r_i}\right)\}. \tag{6}$$

Our model estimates the initial states for each individual. This consists of two parts, first we define the probability that an individual is naïve or non-naïve at the start of the survey given their age and village, then given an individual's naïve status we define the probability that they are in state $S$, $I$, $ID$, or $D$. We assume that each person's history of prior trachoma infection (before the start of the survey) follows a village-dependent Poisson process with rate $\gamma^{v_i}$. The probability that individual $i$ is naïve at the start of the study is then:

$$\pi^i = \exp\left\{-\gamma^{v_i}(a_i + 0.5)\right\}, \tag{7}$$

where $a_i$ is the age of individual $i$, and the additional 0.5 is a continuity correction. We define $\delta_{S0}^{v_i}$, $\delta_{I0}^{v_i}$, $\delta_{ID0}^{v_i}$, and $\delta_{D0}^{v_i}$ as the probability of a naïve individual starting in state $S_0$, $I_0$, $ID_0$, $D_0$ respectively. Likewise, we define $\delta_{S1}^{v_i}$, $\delta_{I1}^{v_i}$, $\delta_{ID1}^{v_i}$, and $\delta_{D1}^{v_i}$ as the probability of a non-naive individual starting in state $S_1$, $I_1$, $ID_1$, $D_1$ respectively. Additionally, ages are missing for fourteen individuals, and so these are included as extra model parameters.

## Likelihood and observation process

We construct a likelihood function to assign individuals to an epidemiological state at each time point based on their diagnostic and clinical observations. We use three types of observation: i) PCR targeting a plasmid sequence, ii) antigen detection tests, and iii) the results of clinical examinations. We use PCR and antigen detection tests to determine *C. trachomatis* infection; whether individuals are in $I$ or $ID$, and clinical examinations indicate TF; whether individuals are in $ID$ or $D$. We estimate the sensitivity ($\phi$) and specificity ($\psi$) of each observation type, assuming these are the same between the two study sites. This gives six diagnostic parameters, see Table 2.

## Inference methodology

We denote $\Theta$ as the model parameters, $\boldsymbol{X}_{1:T}$ as hidden infection states of all individuals for weeks $1,\ldots,T$, and $\boldsymbol{Y}_{1:T}$ as diagnostic and clinical observations. We sample from the posterior distribution

$$p(\Theta, \boldsymbol{X}_{1:T} \mid \boldsymbol{Y}_{1:T}) \propto p(\boldsymbol{Y}_{1:T} \mid \boldsymbol{X}_{1:T}, \Theta)p(\boldsymbol{X}_{1:T} \mid \Theta)p(\Theta) \tag{8}$$

where $p(\Theta)$ is the combined density of the prior distributions, $p(\boldsymbol{X}_{1:T} \mid \Theta)$ is given by the stochastic transmission model, and $p(\boldsymbol{Y}_{1:T} \mid \boldsymbol{X}_{1:T}, \Theta)$ is given by the sensitivity and specificity of the observations. We implement a Markov chain Monte

**Table 2. Prior probability distributions used for diagnostic parameters.**

| Diagnostic | Parameter | Symbol | Prior | Sources |
|---|---|---|---|---|
| PCR | Sensitivity | $\phi_{PCR}$ | beta(5, 2) | [22–25] |
| | Specificity | $\psi_{PCR}$ | beta(20, 1) | [22–25] |
| Antigen detection test | Sensitivity | $\phi_{ADT}$ | beta(5, 2) | [22–25] |
| | Specificity | $\psi_{ADT}$ | beta(20, 1) | [22–25] |
| Clinical eye exam | Sensitivity | $\phi_{CEX}$ | beta(25, 1) | [16,26] |
| | Specificity | $\psi_{CEX}$ | beta(25, 1) | [16,26] |

Carlo (MCMC) algorithm to obtain samples of the posterior distribution, alternating between updating model parameters conditional on the hidden infection states and updating the hidden infection states conditional on the model parameters. The full inference procedure is summarised in S1 Text, with key details outlined below.

Although some model parameters are village dependent, we fit the model to the data from both villages in a single analysis. Combining data from both villages increases the information available for the shared parameters, directly improving the precision of their estimates. Since the village-dependent parameters may depend on these shared quantities, estimates of these parameters may also benefit indirectly from the joint analysis.

When updating the model parameters conditional on the hidden infection states we use a variety of MCMC updates that utilise available conjugacy and provide efficient mixing. For the initial state probabilities, sensitivity, and specificity parameters, our choice of conjugate prior distributions means that the full conditional distributions are available, allowing efficient Gibbs updates. For the unknown age parameters we use independence Metropolis proposals using their prior distributions. We find that there is little information in the data to estimate these ages, meaning that the posterior distributions closely resemble the prior distributions, and so this is a more efficient approach than using random-walk proposals. For the remaining parameters, the full conditional distributions are not analytically available, and so we use random-walk Metropolis updates. Transmission parameters for each village are updated jointly in blocks to reduce computational cost. As the three transmission parameters contribute to the infection probabilities, which are needed to compute the likelihood for these updates, blocking these parameters avoids multiple calculations of each infection probability. The mean durations are re-parametrised to the probability parameters of the negative binomial distributions to improve mixing. The random-walk proposal distributions are constructed using empirical covariance estimates obtained from pilot runs. The proposal covariance matrix for each block of transmission parameters is taken to be the empirical covariance matrix scaled by a factor of $2.38^2 / 3$, following [27]. Likewise, for the remaining parameters the proposal variance is scaled by $2.38^2$, as these are univariate proposals.

In order to update the hidden infection states conditional on the model parameters, we implement the individual forward-filtering backward-sampling algorithm (IFFBS), as previously described [21]. We loop through individuals $i = 1, 2, ..., 1410$ and update the hidden infection states of individual $i$, conditional on the current sample of infection states for all other individuals ($\boldsymbol{X}_{1:T}^{-i}$) and model parameters ($\Theta$):

$$\boldsymbol{X}_{1:T}^i \sim p(\boldsymbol{X}_{1:T}^i \mid \boldsymbol{Y}_{1:T}^i, \boldsymbol{X}_{1:T}^{-i}, \Theta). \tag{9}$$

As negative binomial durations yield a semi-Markov model, this cannot be done directly. Instead, we first construct a Markovian approximation to our stochastic transmission model by setting the size parameters of the negative binomial distributions to 1 (corresponding to geometric distributions) while maintaining the same mean durations. For each individual, we propose a set of hidden infection states using IFFBS with the approximate model, and then apply a Metropolis-Hastings accept-reject step to correct for the discrepancy between the approximate and correct models. This accept-reject

step ensures that the resulting Markov chain has the correct stationary distribution, and posterior samples are drawn from the true semi-Markov model rather than from the Markov approximation. As demonstrated in [21], this is a highly efficient way to sample the hidden states of semi-Markov models, particularly when the size parameters are close to 1, as is the case here.

Particular attention is paid to the durations from the start of the survey. Since our model is non-Markovian, we need to estimate how long each individual has been in their current state, and calculate the remaining duration accordingly. These durations will be upwardly biased due to length-biased sampling [28]. This occurs when the probability that an observation is included in a sample increases with the length (or duration) of the phenomenon being measured. In our setting, the longer the duration of infection, the more likely it is to overlap with the start of the survey. As a result, longer infection durations are overrepresented, and the sample is not representative of the underlying population. For the durations from $I \to ID$, and from $ID \to D$, the corresponding length-biased distributions can be derived analytically [28]. The length-biased distributions from state $D$ are more challenging, as individuals can transition to $S$ via recovery, or to $ID$ via reinfection. Formal derivation of the length-biased durations would therefore require each person's reinfection risk prior to the survey period, which is computationally burdensome to estimate. Instead, for the purpose of approximating the length-biased distribution, we assume a constant value $\alpha^{v_i}$ for all individuals. This means that the durations from $D$ are the smaller of the durations from: a negative binomial distribution governing the transition from $D \to S$ (recovery), and a geometric distribution governing the transition from $D \to ID$ (reinfection). Under this simplification we can determine the length-biased duration distributions from state $D$ numerically.

## Prior distributions

We use informative prior distributions for the observation sensitivities and specificities based on previous studies (Table 2). Although informative prior distributions can be constructed for the PCR tests from previous studies, comparable data are not available for antigen detection tests in this setting, and we therefore use the same prior distributions for these two diagnostics. This choice reflects prior uncertainty about the diagnostic performance in this setting, while avoiding favouring one test over another *a priori*. For the remaining model parameters, we use weakly informative prior distributions to constrain the parameters within sensible ranges.

For the six transmission parameters, we use exponential priors with rate 5, allowing transmission probabilities to be small or large. E.g., if we consider a single susceptible sharing a village with a single infected, the 95% prior HDR credible interval for the weekly infection probability would be (0.00, 0.45). For the means of the durations, we re-parameterise to the probability parameter of the negative binomial distribution, and use a uniform prior distribution on the range (0, 1). We constrain $\mu_{ID0}$ and $\mu_{D0}$ to be larger than $\mu_{ID1}$ and $\mu_{D1}$ respectively, as naïve infections take longer to clear on average. For the relative reinfection risk parameter $\rho$ we use a $\Gamma(2, 0.5)$ distribution, allowing for the infection risk to be lower or greater for a diseased individual compared to a susceptible individual. The parameters $\alpha^{v_i}$ used to approximate the length-biased distribution from state $D$ are given an exponential prior distributions with rate 20, which translates to a 95% prior HDR credible interval of (0.00, 0.15).

For historic infection rates $\gamma^{v_i}$ we use a $\Gamma(1.5, 1/0.75)$ distribution, which broadly covers 1 infection every 6 years to 6 infections per year. The prior distributions for the probability vectors $(\delta_{S0}^{v_i}, \delta_{I0}^{v_i}, \delta_{ID0}^{v_i}, \delta_{D0}^{v_i})$ and $(\delta_{S1}^{v_i}, \delta_{I1}^{v_i}, \delta_{ID1}^{v_i}, \delta_{D1}^{v_i})$ are non-informative Dirichlet distributions with all concentration parameters equal to 0.5.

A total of 14 individuals have unknown ages, and so these unknown ages are included as additional parameters. The prior distribution for each age parameter follows the empirical age distribution of the relevant village.

## Model fit

We ran four parallel MCMC chains for 150,000 iterations, which took approximately 150 hours on a 3 GHz processor. We examined the posterior chains and obtained a multivariate Gelman-Rubin statistic of 1.01, indicating good convergence.

## Results

### Prevalence of *C. trachomatis* and ocular disease

We fit the epidemiological model (described in Methods) to cohort data on trachoma infection from two villages in West Africa (Table 1). We start by inferring the prevalence of infection with *C. trachomatis*: $(N_I + N_{ID})/N_{tot}$; and TF: $(N_{ID} + N_D)/N_{tot}$, where $N_I$, $N_{ID}$, $N_D$, and $N_{tot}$ are the number of individuals in state *I*, state *ID*, state *D*, and the total number of individuals respectively. These numbers are derived through time from the inference of hidden epidemiological states for each individual (an example is shown in Fig 2). Both villages were endemic for trachoma during the study period, see Fig 3. In the larger village (*J*), the prevalence of infection with *C. trachomatis* started at 18%, declining to 12% by the end of the observed period. In the second village (*B*) the prevalence of infection was more stable at around 11% over the observed period. The prevalence of TF was higher than active infection in both villages, exceeding 20% in the larger village (*J*) for most of the study period, while the prevalence of TF was around 16% in the second village (*B*).

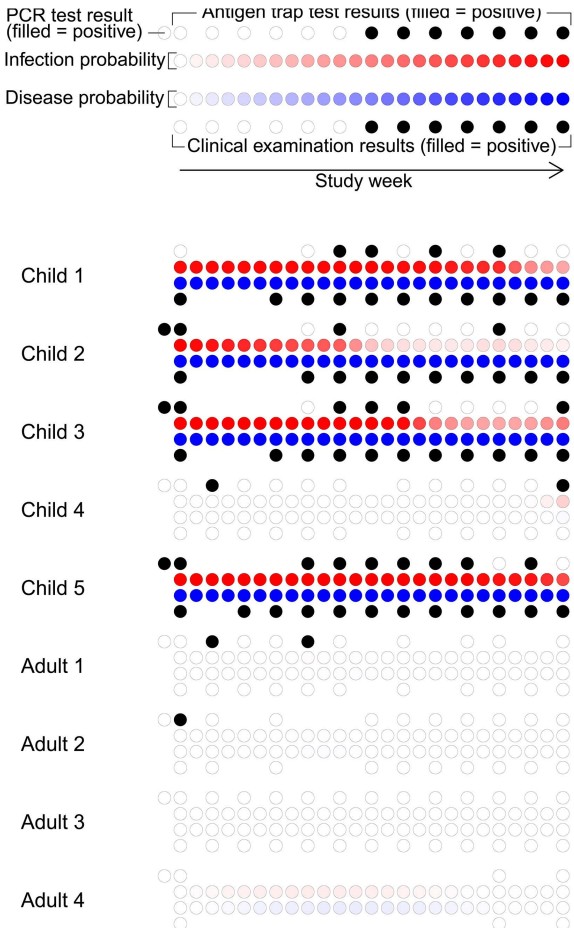

**Fig 2. Example of state posterior estimates for nine people sharing a room.** Each person is represented by four rows of circles. The first row shows antigen detection test results (hollow for negative, black solid for positive), plus the results of the baseline PCR tests offset to the left of the baseline antigen detection test result. Missing circles indicate missing data. The second row shows 25 circles indicating the marginal probability of the person being infected (*I* or *ID* states) for each week. Darker shades indicate higher probabilities. The third row shows the marginal posterior probabilities of being diseased (*ID* or *D* states). Finally, the fourth row shows the results of clinical examinations for ocular disease.

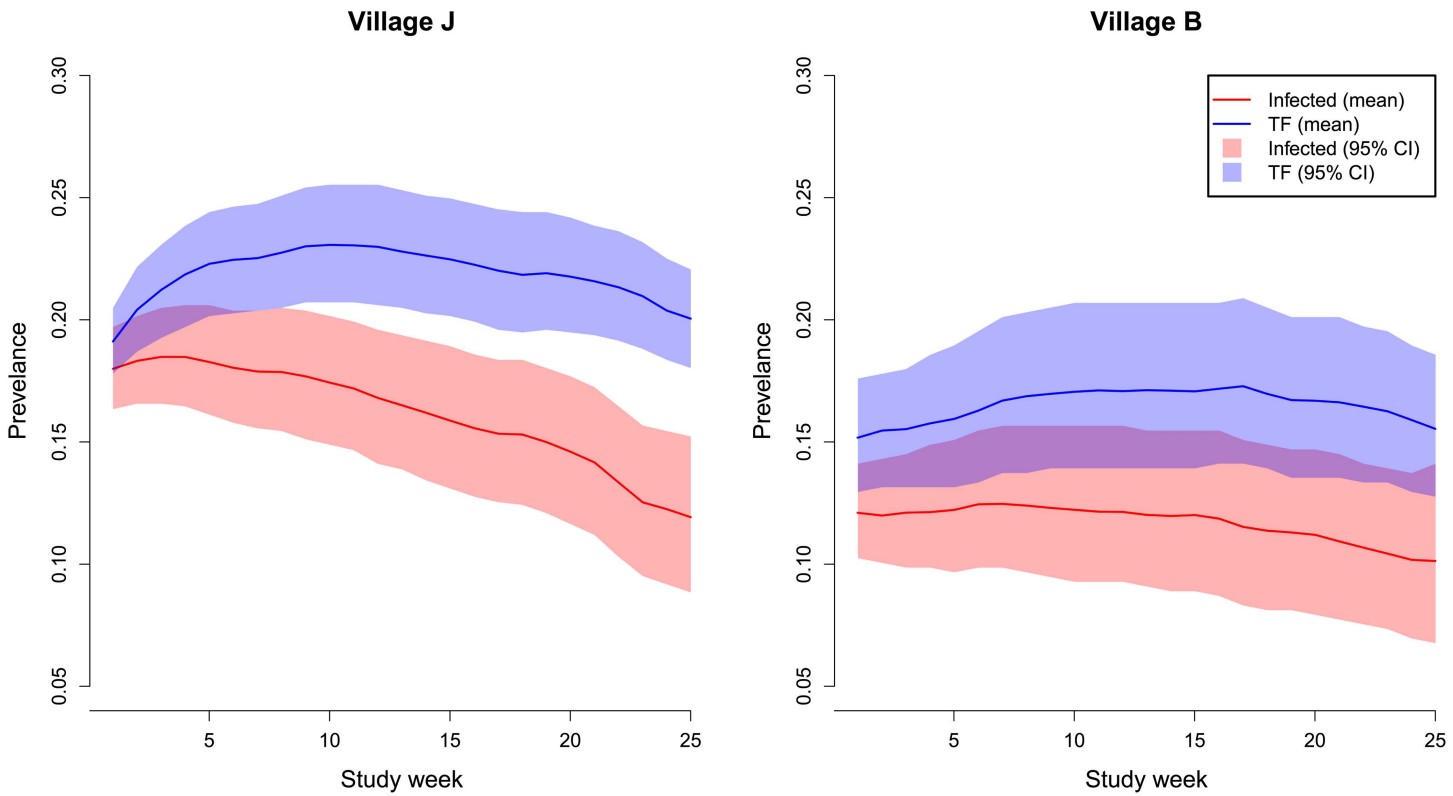

**Fig 3. Prevalence of conjunctival Chlamydia trachomatis infection and trachomatous inflammation–follicular (TF) in two villages in The Gambia over 25 weeks, as inferred by a hidden state epidemiological model.**

## Transmission of trachoma

Our results indicate that the probability of infection with *C. trachomatis* is strongly affected by the proximity and nature of contact in domestic settings. The posterior mean probability of a susceptible person becoming infected ($S \rightarrow I$) given one infectious case residing within the same room is 2.1 $\times 10^{-3}$ (95% credible interval [CrI] 3.2 $\times 10^{-4}$, 4.8 $\times 10^{-3}$) per week, which is 39-fold greater than the mean probability of transmission given one infected person residing within the same village (5.4 $\times 10^{-5}$, 95% CrI 2.4 $\times 10^{-5}$, 9.1 $\times 10^{-5}$). These values are from the larger village (J), though the effect is even greater in the second village (B) where the probability of transmission increases 48-fold in a shared room compared to a shared village, see Table 3.

We assess whether individuals who have previously been infected and have ocular disease (*D*) are more likely to become reinfected than susceptible individuals (*S*). The posterior distribution of the reinfection parameter $\rho$ is broad, with a mean of 1.24 (95% CrI 0.27, 2.79), though the corresponding effect on infection risk is non-linear due to the exponential function (Eq 6). While the posterior mean exceeds 1, suggesting higher susceptibility among individuals with TF in contrast to earlier analyses [29], this estimate is highly uncertain. Notably, 41% of the posterior mass lies below 1, indicating substantial support for the alternative possibility that individuals in the *D* state are less susceptible than those in the *S* state. We observe no strong posterior correlation between $\rho$ and other model parameters, indicating that this uncertainty is not driven by parameter confounding but rather by limited information in the data. Although there are plausible biological mechanisms that could increase infection risk among diseased individuals (e.g., increased eye touching or eyelid

PLOS Computational Biology

**Table 3. Risk of a susceptible person acquiring conjunctival Chlamydial trachomatis infection with trachoma per week, both as probabilities ($p_t^i$, see Equation 1) and relative risk, given one infectious case in a shared village, compound, or room.**

| Location | Risk type | Village | Compound | Room |
|---|---|---|---|---|
| Jali (J) | Probability | $5.4 \times 10^{-5}$ | $2.2 \times 10^{-4}$ | $2.1 \times 10^{-3}$ |
| | Relative | 1 | 4.0 | 39 |
| Berending (B) | Probability | $5.8 \times 10^{-5}$ | $1.8 \times 10^{-3}$ | $2.8 \times 10^{-3}$ |
| | Relative | 1 | 31 | 48 |

damage), the data do not strongly constrain $\rho$, and conclusions regarding differences in susceptibility should therefore be interpreted with caution.

## Basic reproduction number

We estimate the basic reproduction number ($R_0$) for conjunctival *C. trachomatis* and the distribution of secondary cases by simulation. We repeatedly sample a set of parameters from the posterior distribution, as well as an individual to act as the initial infected among a susceptible population. For this individual, we simulate their total infectious durations (progressing from $I$ to $D$, assuming they are naïve), and the number of individuals they infect over this period. Averaging over a large number of samples provides an estimate of $R_0$. We estimate $R_0$ for the largest village ($J$) as 2.2 and for the second village ($B$) as 2.4.

The simulated distribution of secondary cases is shown in Fig 4. Fitting the negative binomial distribution to the secondary cases we obtain a value for the size parameter $k$ as 1.81 for village $J$ and 1.61 for village $B$. A small value of $k$ (e.g., < 1) indicates highly heterogenous transmission, with most secondary cases arising from a small number of individuals ("superspreaders"). As $k$ increases, transmission becomes more evenly distributed across individuals, and the secondary case distribution converges towards a Poisson distribution. A previous study estimated the values of $k$ for several past outbreaks (including SARS, measles, monkeypox, and pneumonic plague) to be in the range of $0.01 - 1$ [30]. Our results therefore indicate a lower dispersion of secondary cases compared with other directly transmitted pathogens.

## Heterogeneity in susceptibility within the cohort

The mean posterior probability of becoming infected ($S \rightarrow I$, given by $p_t^i$) per week is 0.0098 across both villages, which reflects the continuous infection pressure in both sites throughout the observed period. The distribution of these probabilities for the 1,410 individuals in the cohort shows considerable heterogeneity per week, ranging 9-fold from 0.0031 to 0.028, which is consistent with the parametric distribution beta(6, 600) fitted by maximum likelihood to the posterior output, see Fig 5. The cumulative probability of becoming infected in week $t$ for person $i$ is defined as $Cp_t^i = 1 - \prod_{k=1}^{t}(1 - p_k^i)$. This rises to a mean of 0.22 over 25 weeks of exposure and the cumulative individual risk at the end of the observed period varies 5-fold from 0.085 to 0.42 (Fig 5). We quantify this variance in individual infection risk by fitting the parametric model $Cp_t^i = 1 - e^{-\zeta \cdot g_i \cdot t}$; $\zeta$ gives the intercept, $t$ the study week, and $g_i$ represents individual-level predisposition to infection, where $g \sim \text{Gamma}(\omega, \omega)$. Through Bayesian inference, we obtain mean posteriors as $\zeta = 0.0103$ and $\omega = 6.40$. This fitted function is shown as the dashed line in (Fig 5).

## Contribution of age groups to transmission

We examine the relative contribution of different age groups to the weekly probability of trachoma transmission, given by the force of infection (Eq 1). We find that young children (0–9 years) contribute disproportionately to transmission, accounting for more than 75% of the infection pressure in the largest village ($J$) and more than 70% in the second village ($B$) over the observed period, despite accounting for less than half of the population ($J$: 36%, $B$: 40%), see Fig 6. Adults

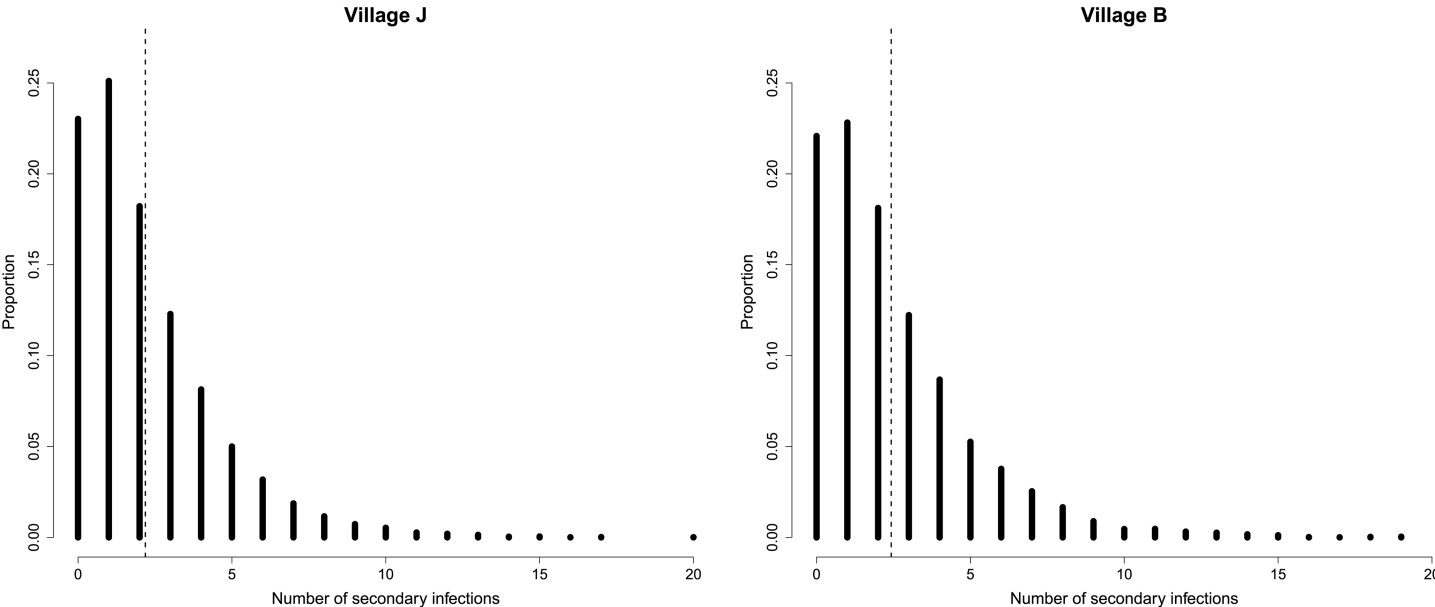

**Fig 4. Estimated distribution of secondary cases from a single infected in a fully susceptible population, inferred by simulation, and the basic reproduction number (mean of distribution; dashed line).**

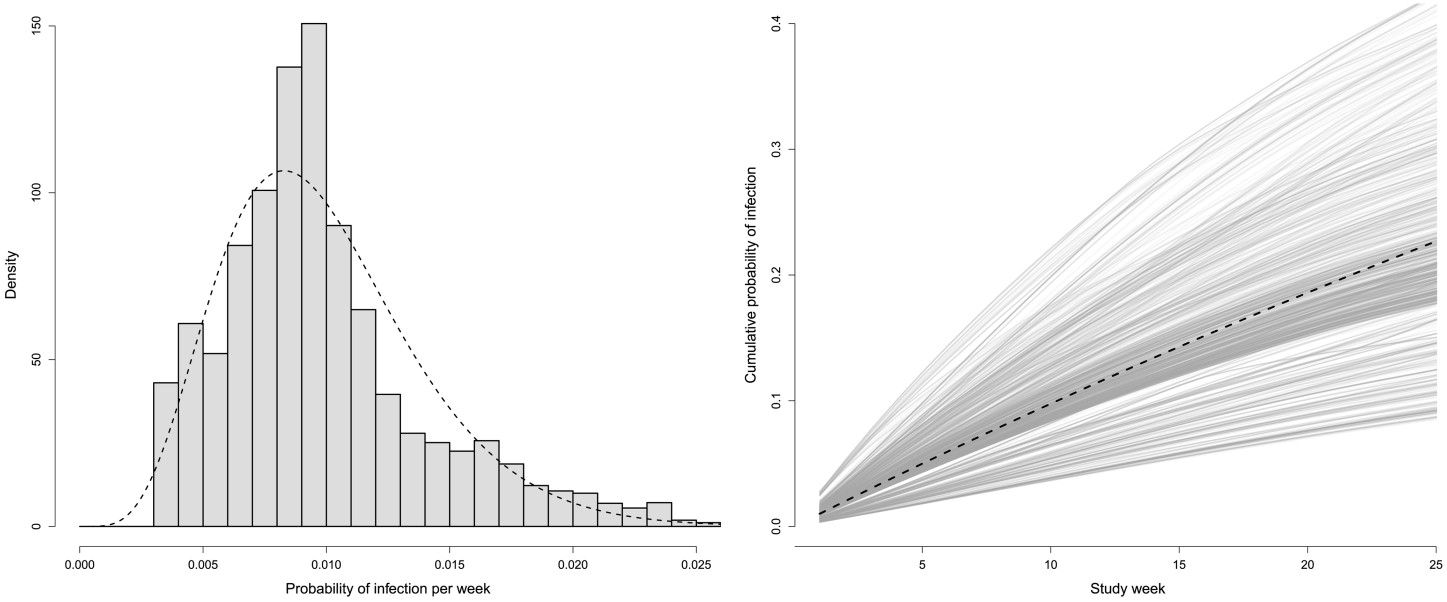

**Fig 5. Heterogeneity in trachoma susceptibility.** Left: The distribution of the weekly posterior probability of becoming infected as a histogram, with an overlaid parametric distribution fitted by maximum likelihood; beta(6, 600). Right: The cumulative posterior probability of infection ($Cp_t^i$) for all 1,410 individuals ($i$) over the 25 weeks ($t$) of the cohort study, where *the* dashed line gives the posterior mean of the fitted function $1 - exp(-\zeta \cdot g \cdot t)$, where $\zeta = 0.0103$ and $g \sim gamma(6.4, 6.4)$.

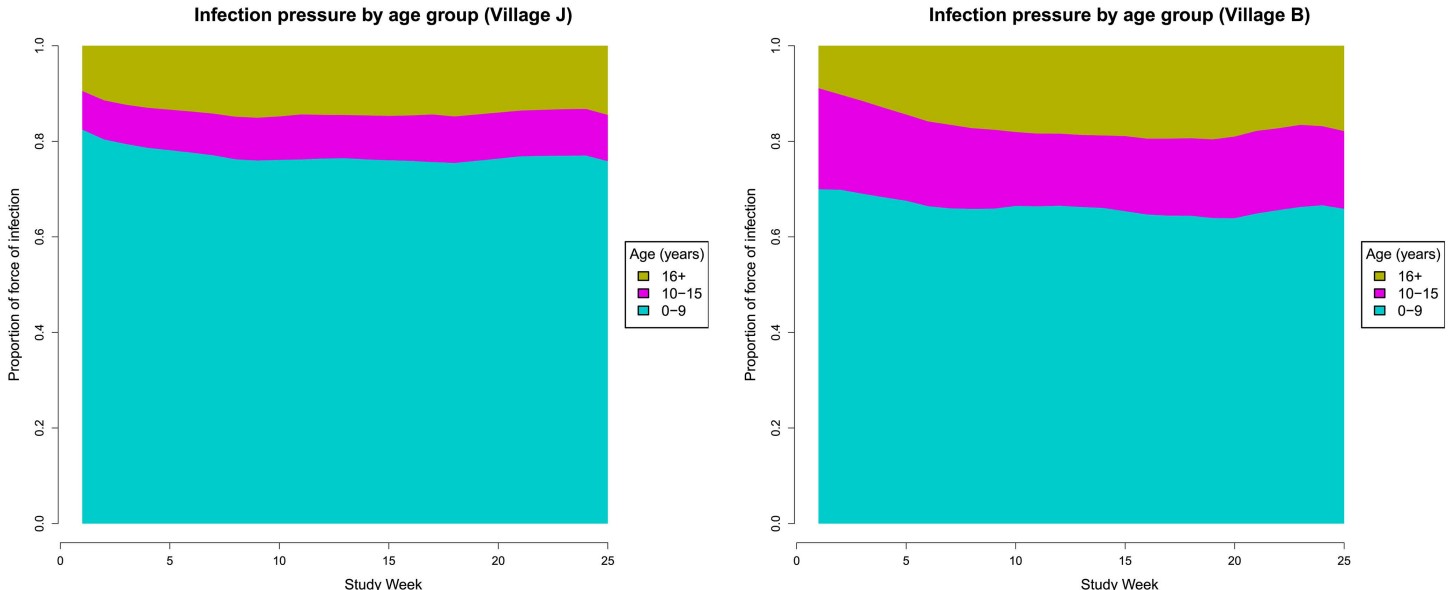

**Fig 6. Relative contribution of age groups to the force of infection (Eq 1) with each panel representing a village (Jali [*J*] and Berending [*B*]) in The Gambia (Table 1).**

16 + years of age contribute under 20% to the force of infection in both villages, which is a consequence of having a lower duration of infectiousness due to past infections.

## Observation sensitivities and specificities

We obtain posterior means (95% credible intervals) for the specificities $\psi_{PCR}$ : 0.95 (0.93, 0.96), $\psi_{ADT}$ : 0.95 (0.94, 0.96), $\psi_{CEX}$ : 0.98 (0.97, 0.98), and sensitivities $\phi_{PCR}$ : 0.94 (0.88, 0.98), $\phi_{ADT}$ : 0.74 (0.69, 0.79), $\phi_{CEX}$ : 0.91 (0.88, 0.94). The marginal posterior distributions are shown in Fig 7 along with each prior distribution. In general, we learn a lot about the parameters, but obtain more confident estimates for the specificity parameters compared to the sensitivity parameters.

For the PCR test, the specificity posterior mean is similar to the prior, but the sensitivity estimates are at the higher end of previously published values [22]. We determine that the specificity of the antigen detection test is very similar to that of PCR, but the sensitivity is significantly lower. For the clinical examinations, both the specificity and sensitivity were high.

## Initial infection probabilities

The probability that an individual starts in any of the eight infection states has two components. First, the probability of being classed as naïve at the start of the survey (depending on age and village), determined by the historic yearly infection rates $\gamma^B$ and $\gamma^J$ for villages $B$ and $J$ respectively. Second, conditional on an individual's village and naïve status, the probability of being in state $S$, $I$, $ID$, or $D$, giving 16 parameters $\delta_z^v$ for $v \in \{B, J\}$ and $z \in \{S_0, I_0, ID_0, D_0, S_1, I_1, ID_1, D_1\}$.

The posterior mean (95% credible interval) for $\gamma^J$ is 0.14 (0.11, 0.18) and for $\gamma^B$ is 0.11 (0.07, 0.16). These values suggest that the historic infection rate is slightly larger in village $J$ compared to village $B$. In Fig 8 we compare the probabilities that individuals of a given age are classed as naïve at the start of the survey. Most people appear to have experienced at least one infection in early childhood in both villages. However, we should be careful not to over-interpret these parameters, as it is possible that some non-naïve individuals exhibit longer than anticipated recovery durations and are thus categorised as naïve in the model (or vice versa).

**Fig 7. Marginal posterior distributions for the specificities and sensitivities.** Histograms show the posterior distributions, and the lines show the prior distributions.

For naïve individuals in village *J*, the probability of starting in state *S*, *I*, *ID*, and *D* is estimated from the posterior mean to be 0.25, 0.01, 0.61, and 0.13 respectively. For non-naïve individuals these probabilities are 0.96, 0.02, 0.01, and 0.01. For village *B*, the naïve probabilities are 0.55, 0.01, 0.32, and 0.12, and the non-naïve probabilities are 0.96, 0.01, 0.03, and 0.00. These probabilities relate to the durations for each state as naïve individuals take longer to progress from the *ID* state than non-naïve individuals and so are more likely to be sampled in this state. The probability that a naïve individual in village *J* starts in the *S* state is much lower than village *B* (Fig 8), further supporting the notion that the historic infection rate is higher in village *J*.

## Durations in epidemiological states

The posterior mean (95% credible intervals) for the duration parameters are presented in Table 4. As expected, there is a significant reduction in the clearance and recovery durations for non-naïve individuals.

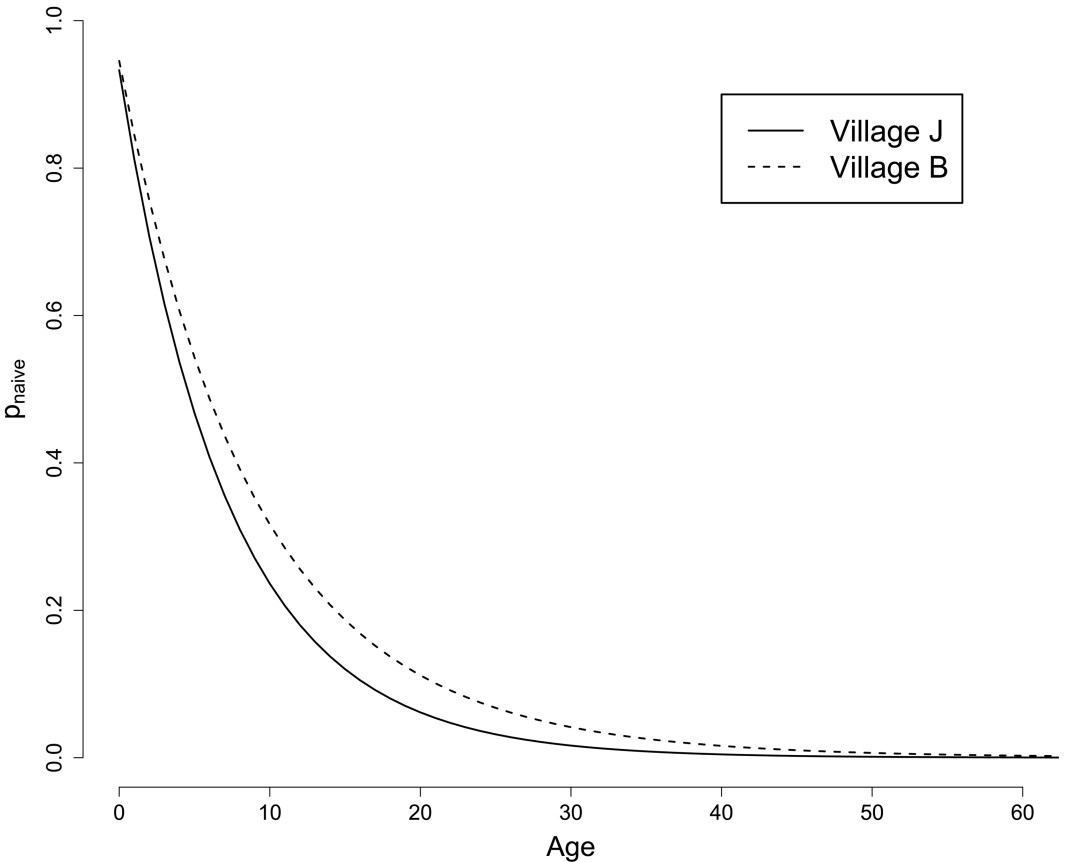

**Fig 8. Age-dependent probability that an individual has no previous infections at the start of the observed period.**

**Table 4. Posterior means and 95% credible intervals for the duration parameters in weeks.** $\mu_I$ corresponds to the infectious states ($I_0$ and $I_1$), $\mu_{ID0}$ and $\mu_{ID1}$ correspond to the naive and non-naive infectious and diseased states ($ID_0$ and $ID_1$), and $\mu_{D0}$ and $\mu_{D1}$ correspond to the naive and non-naive diseased states ($D_0$ and $D_1$).

| Parameter | Posterior mean | 95% credible interval |
|---|---|---|
| $\mu_I$ | 1.6 | (1.0, 3.9) |
| $\mu_{ID0}$ | 35 | (24, 52) |
| $\mu_{ID1}$ | 4.7 | (2.3, 8.1) |
| $\mu_{D0}$ | 36 | (20, 66) |
| $\mu_{D1}$ | 1.8 | (1.0, 3.1) |

Individual durations are stochastic, following a negative binomial distribution with size parameter 2, and so there is large variation. Naïve individuals clear their first infection (progress from $I_0$ to $D_0$; mean (95% credible interval)) in 37 (5, 104) weeks, and non-naïve individuals clear infections (progress from $I_1$ to $D_1$) in 6.2 (2, 16) weeks. The estimates of $\mu_{D0}$ and $\mu_{D1}$ may not be indicative of individual durations from $D$ to $S$, due to the possibility of reinfection. We simulate realisations of individual durations from $D$ to $S$ by randomly selecting individuals and using their average weekly infection probability as a constant weekly reinfection probability, leading to geometrically distributed reinfection durations. For naïve

individuals we obtain recovery durations (progress from $D_0$ to $S_1$) of 28 (4, 83) weeks, and for non-naïve individuals we obtain recovery durations (progress from $D_1$ to $S_1$) of 1.8 (1, 5) weeks.

## Discussion

We have developed a Bayesian inference framework to infer hidden epidemiological states from longitudinal observations and applied this to understanding the natural history of trachoma infection from a unique cohort in West Africa. Our statistical framework combines detailed data on the structure of the population (organised into rooms, compounds and villages), participants' age, and (imperfect) diagnostic test results to calculate a posterior probability of infection for each individual in each week of the study period. Such a detailed, individual-level analysis provides unique insights into the precise epidemiology of infection in this population.

Our key findings are that the basic reproduction number $R_0$ for *C. trachomatis* is estimated as 2.2 and 2.4 respectively in the two study villages, and the force of infection is disproportionately driven by children ≤9 years of age (Figs 4 and 6). These results are supported by the finding that children 1–9 years old are disproportionately likely to have active trachoma and ocular *C. trachomatis*, as determined by quantitative PCR, in a study in Ethiopia [31]. There are relatively few estimates of $R_0$ for trachoma, owing to a lack of longitudinal studies, though a previous analysis estimated $R_0$ in children 1–9 years old using serological data from Tanzania [32], finding that $R_0$ ranged between 2.8–28 across three types of settings. Our $R_0$ results are consistent with the results from hypoendemic (low) transmission settings (95% confidence interval 1.6–4.0 [32]), though our simulations use the full age distribution within communities, including adults who have lower transmission rates.

The household nature of trachoma transmission is known from epidemiological studies in the 1980s and 1990s [33], however we are the first to quantify this effect by estimating the probability of infection as a function of the number of infectious individuals within the same room, compound or village.

We are the first to simulate the distribution of secondary cases, and we find that relatively few primary cases (around 24%) result in zero subsequent infections. This relatively low heterogeneity reflects the long infectious period for trachoma and the household structure (Table 1), whereby multiple people sharing rooms causes transmission events (Table 3).

Our results suggest that transmission-blocking interventions targeted at young children could substantially reduce trachoma transmission within communities. Current elimination programmes are largely based on the SAFE strategy (surgery for trichiasis, antibiotics, facial cleanliness, and environmental improvement), which encompasses WASH (water, sanitation, and hygiene) [8,34]. Annual mass treatment with azithromycin remains the cornerstone of control efforts, and has proven effective at reducing trachoma infection and the resulting blindness [3,6]. While observational studies have reported associations between household water access or facial cleanliness and lower odds of active trachoma [35,36], randomised trials in Ethiopia have not demonstrated a measurable impact of WASH provision on trachoma prevalence [10,37]. Despite major progress towards elimination, persistent endemic foci remain, most notably in Ethiopia, prompting renewed interest in novel interventions such as vaccines against Chlamydia trachomatis [38]. A candidate antigen, CTH522, is currently in clinical trials [39], and our findings add to the evidence base supporting the prioritisation of transmission-blocking vaccines targeted at children.

This study also has several important limitations. Transmission events are not observed directly, but are instead inferred from longitudinal diagnostic and clinical observations by fitting a stochastic transmission model. While the availability of repeated measurements alongside the detailed household structure of the population allows us to infer transmission dynamics with greater clarity than cross-sectional data, we still can not unambiguously identify who-infected-whom. Furthermore, we lack complementary data such as pathogen genome sequences that could provide circumstantial evidence of direct transmission between individuals. The transmission parameters therefore represent average effects at the level of shared environments, rather than confirmed person-to-person transmission events. Future studies combining dense longitudinal data with pathogen genomic data could help resolve transmission links more precisely, and further validate model-based inference.

 

Overall, this study demonstrates how a principled statistical approach applied to detailed longitudinal data can be used to jointly infer hidden infection states and key natural history parameters, yielding new insights into the underlying dynamics of trachoma infection.

## Supporting information

**S1 Text. Algorithmic description of the Bayesian inference procedure.** Summary of the Markov chain Monte Carlo scheme and hidden infection state sampling (Algorithm S1).
(PDF)

## Author contributions

**Conceptualization:** Robin Bailey, T. Déirdre Hollingsworth, Simon E. F. Spencer.

**Data curation:** Robin Bailey.

**Formal analysis:** Jake Carson, Thomas Crellen, Anna Borlase, Joaquin M Prada.

**Funding acquisition:** T. Déirdre Hollingsworth, Simon E. F. Spencer.

**Methodology:** Jake Carson, Simon E. F. Spencer.

**Writing – original draft:** Jake Carson, Simon E. F. Spencer.

**Writing – review & editing:** Thomas Crellen, Anna Borlase, Joaquin M Prada, T. Déirdre Hollingsworth.

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
