## [Decision Letter · Decision Letter 0]

12 Mar 2026

PCOMPBIOL-D-25-02394

Dynamics of trachoma infection in West Africa revealed by a hidden state model

PLOS Computational Biology

Dear Dr. Carson,

Thank you for submitting your manuscript to PLOS Computational Biology. After careful consideration, we feel that it has merit but does not fully meet PLOS Computational Biology's publication criteria as it currently stands. Therefore, we invite you to submit a revised version of the manuscript that addresses the points raised during the review process. The reviewers are incredibly positive about your work, but there are some missing methodological details that are required (see reviewer 3).

We look forward to receiving your revised manuscript.

Kind regards,

Oliver Eales

Academic Editor

PLOS Computational Biology

Benjamin Althouse

Section Editor

PLOS Computational Biology

**Journal Requirements:**

3) Please amend your detailed Financial Disclosure statement. This is published with the article. It must therefore be completed in full sentences and contain the exact wording you wish to be published.

4) Please ensure that the funders and grant numbers match between the Financial Disclosure field and the Funding Information tab in your submission form. Note that the funders must be provided in the same order in both places as well. Currently, the order of the grants is different in both places.

**Reviewers' comments:**

Reviewer's Responses to Questions

**Comments to the Authors:**

**Please note that one review is uploaded as an attachment.**

Reviewer #1: Thank you for this important paper

I have minimal suggestions for improvements

In the section Traversal times you might consider renaming as durations in disease states and provide results as a table with the symbols also explained in words

Results - you present the findings separately for each village which is helpful but I wondered how you might consider combining the data from two or more villages to potentially give more accurate estimates of the key biological parameters?

Discussion highlights clearly the implications of the findings. R0 remains >2 which presents significant challenges for control when reinfection is common. Are there other harm reduction methods alongside vaccines e.g. related to WASH or other specific practices within the home?

I was able to locate the code and supplementary files at zenodo although i have not reviewed these in detail

Overall I found this paper super interesting and a great application of the methods to this ongoing challenge

I look forward to seeing the method applied to other infections to combined multiple data sources

Reviewer #2: This paper presents a very nice re-analysis of unique data, and offers a variety of intriguing outputs that are unlikely to be estimated based on empirical data in any other way. For that reason it’s a technically significant manuscript. The authorship team is exemplary. My comments are all very pedantic.

Line 11: rather than “currently” suggest specify a date – this paper will be read and used for a while to come

Line 16: please use an em-dash between “inflammation” and “follicular”, as specified in the report of the 4th Global Scientific Meeting on Trachoma. You’ve written it correctly in line 86, but at that point the abbreviation has already been defined

Line 20: please change “face washing” to “facial cleanliness”

Line 24: it would be preferable to refer to programmes as “trachoma elimination programmes” – this speaks to the agreed global public health target

Line 62 (and subsequently): I don’t think “plasmid PCR” is quite right. “PCR targeting a plasmid sequence” is probably ok. Similarly, I’ve not previously referred to assays as “antigen trap test”; “antigen detection test” would be better.

Line 77: please edit “one 1’s”; if you retain the digit rather than the word, it doesn’t need an apostrophe.

Line 158: TF means a very specific thing. Please don’t say “follicular disease (TF)”: not all follicular conjunctivitis meets the criteria for TF. Here TF is intended, so just say “TF”.

Line 235 and 243: please change “follicular trachoma” and “ocular disease (TF)” here to “TF”

Caption Figure 3: Please change “Prevalence of infection with Chlamydia trachomatis (I) and ocular disease with follicular trachoma” to “Prevalence of conjunctival Chlamydia trachomatis infection and trachomatous inflammation—follicular (TF)”. Please alter the inset box in the figure area to change “diseased” to “TF” and “Infectious” to “Infected”: there is other disease from trachoma other than TF, and “infected” and “infectious” are not necessarily the same thing.

Caption Table 3: please change “becoming infected with trachoma” to “acquiring conjunctival Chlamydial trachomatis infection”

Line 259: abbreviation already defined

Line 268: here I think it’s important to qualify that the estimate is of the R0 for “conjunctival” C. trachomatis.

Figure 5 caption: please check that “infectiousness” is really what is being represented here.

Line 296: should 0–9 years be 1–9 years? See Figure 6.

Suggest add a limitations para to the discussion. This should include the fact that actual transmission events were not observed; nor were they inferable by, for example, sequencing isolates to derived circumstantial evidence of transmission between people sharing rooms.

Reviewer #3: Uploaded as an attachment.

**Have the authors made all data and (if applicable) computational code underlying the findings in their manuscript fully available?**

Reviewer #1: Yes

Reviewer #2: Yes

Reviewer #3: Yes

PLOS authors have the option to publish the peer review history of their article (what does this mean?). If published, this will include your full peer review and any attached files.

Reviewer #1: No

Reviewer #2: No

Reviewer #3: No

**Figure resubmission:**
---

## [Decision Letter · Decision Letter 1]

11 May 2026

Dear Dr Carson,

We are pleased to inform you that your manuscript 'Dynamics of trachoma infection in West Africa revealed by a hidden state model' has been provisionally accepted for publication in PLOS Computational Biology.

Best regards,

Oliver Eales

Academic Editor

PLOS Computational Biology

Benjamin Althouse

Section Editor

PLOS Computational Biology

Reviewer's Responses to Questions

**Comments to the Authors:**

Reviewer #2: I remain impressed with the scholarship and presentation. My comments on the original submission have been very well addressed.

Reviewer #3: The authors have made changes addressing all of my concerns and I would be happy to see the work published.

**Have the authors made all data and (if applicable) computational code underlying the findings in their manuscript fully available?**

Reviewer #2: Yes

Reviewer #3: Yes

PLOS authors have the option to publish the peer review history of their article (what does this mean?). If published, this will include your full peer review and any attached files.

Reviewer #2: No

Reviewer #3: No

---

## [Editor Report · Acceptance letter]

PCOMPBIOL-D-25-02394R1

Dynamics of trachoma infection in West Africa revealed by a hidden state model

Dear Dr Carson,

I am pleased to inform you that your manuscript has been formally accepted for publication in PLOS Computational Biology. Your manuscript is now with our production department and you will be notified of the publication date in due course.

With kind regards,

Judit Kozma
